# Biomechanics of the Human Middle Ear with Viscoelasticity of the Maxwell and the Kelvin–Voigt Type and Relaxation Effect

**DOI:** 10.3390/ma13173779

**Published:** 2020-08-27

**Authors:** Rafal Rusinek, Marcin Szymanski, Robert Zablotni

**Affiliations:** 1Department of Applied Mechanics, Mechanical Engineering Faculty, Lublin University of Technology, 20-618 Lublin, Poland; zablotni.robert@gmail.com; 2Department of Otolaryngology Head and Neck Surgery, Medical University of Lublin, 20-059 Lublin, Poland; marcinszym@poczta.onet.pl

**Keywords:** middle ear, stapes vibration, relaxation, ligaments modelling

## Abstract

The middle ear is one of the smallest biomechanical systems in the human body and is responsible for the hearing process. Hearing is modelled in different ways and by various methods. In this paper, three-degree-of-freedom models of the human middle ear with different viscoelastic properties are proposed. Model 1 uses the Maxwell type viscoelasticity, Model 2 is based on the Kelvin–Voigt viscoelasticity, and Model 3 uses the Kelvin–Voigt viscoelasticity with relaxation effect. The primary aim of the study is to compare the models and their dynamic responses to a voice excitation. The novelty of this study lies in using different models of viscoelasticity and relaxation effect that has been previously unstudied. First, mathematical models of the middle ear were built, then they were solved numerically by the Runge–Kutta procedure and finally, numerical results were compared with those obtained from experiments carried out on the temporal bone with the Laser Doppler Vibrometer. The models exhibit differences in the natural frequency and amplitudes near the second resonance. All analysed models can be used for modelling the rapidly changing processes that occur in the ear and to control active middle ear implants.

## 1. Introduction

Hearing is one of the human senses that is very important for communication. During the hearing process, a sound is transferred through the external and middle ear to the cochlea. This process can be described as follows: when sound waves reach the tympanic membrane, they cause the tympanic membrane and the attached chain of auditory ossicles to vibrate. The motion of the stapes against the oval window sets up waves in the fluids of the cochlea, causing displacement of the basilar membrane. Consequently, the sensory cells of the Corti organ are stimulated and send nerve impulses to the brain. From a biomechanical point of view, the middle ear part is the most interesting; therefore, scientists have been looking for methods that allow us to analyse the hearing process correctly. Different models of the human middle ear have been built for decades. The first study in this field was published in 1961 by Mőller [1], where the new scheme of the middle ear mechanism was proposed. Afterwards, a similar model was investigated by Zwislocki [2]. They both based their studies on the popular theory by Bárány, according to which the ossicles rotate about the axis passing through the head of the malleus and the short process of the incus; Zwislocki assumed a rigid coupling between the malleus and the incus. In recent decades, mechanical models of lumped masses have been developed, in which the ossicles (i.e., the malleus, the incus, and the stapes) are represented by rigid bodies, connected by springs and dashpots. The stapes motion for low excitation frequencies, responsible for speech recognition, is piston-like [3]; therefore, usually planar motion is taken into consideration in models reported in the literature. One can find three [4], four [5], and even six [6] degrees of freedom (dof) models of the middle ear (ME). In most cases, these models only focus on the kinematic relations in the intact middle ear. Feng and Gan [6] analysed a dynamic aspect of the middle ear. Their numerical analyses have been in quite good agreement with experimental findings. All the above-mentioned models are linear where joints and ligaments are modelled as the Kelvin–Voigt (K-V) elements. This is a simplification of the real ear because ligaments, joints, and tendons are characterized in the literature as elements with nonlinear behaviour. Generally, the viscoelastic properties of ligaments and tendons in the human body have been widely studied by static creep and stress relaxation experiments [7]. In static creep experiments, the stress is maintained as a constant and the strain is observed to increase over time until it becomes almost constant [8]. It has been demonstrated by static creep experiments that the time needed for the strain to achieve a steady value for different stress levels and different types of ligaments and tendons differs [9]. In static stress relaxation tests on the other hand, the strain is maintained constant and the stress is observed to decrease over time until it finally reaches a steady value. Relaxation experiments at different strain levels of different types of ligaments and tendons have different relaxation times [10,11]; however, the creep and relaxation observed in experimental research are often omitted in modelling of the middle ear. The creep and relaxation phenomena can be modelled in a simplified form as a combination of linear springs and dashpots put together. To develop mathematical models of their viscoelastic behaviour, the K-V and the Maxwell models are used. In the Maxwell model, the spring and the dashpot are in series, while in the Kelvin–Voigt model, they are parallel. The K-V model is usually employed to model the creep phenomenon, whereas the Maxwell model is used for modelling relaxation.

Nonlinear properties of tendons and ligaments in the middle ear are two separate problems that should be discussed here, as nonlinearities can significantly influence system dynamics. The problem has been mentioned in several papers, e.g., [12,13,14,15,16]. For example, Cheng and Gan investigated the nonlinear mechanical properties of the anterior malleolar ligament (AML) via experimental measurement and the material modelling point of view [17]. The hyperelastic Ogden model was used to describe the constitutive behaviour of the annular ligament (AL) in [18]. The authors have proved that the human AL is a typical viscoelastic material with nonlinear stress–strain relationship and stress relaxation function. They found that the transmission of sound energy from the middle ear to the cochlear fluid is largely dependent on the AL mechanical behaviour. As reported in [17,19,20], the mechanical properties of the tensor tympani tendon, the stapedial tendon, and the anterior malleolar ligament (AML) should be described by viscoelastic materials with stress relaxation behaviour.

Thus, the above-mentioned studies demonstrate that joints, tendons, and ligaments are crucial for sound transfer through the middle ear and also for modelling its biomechanical behaviour; therefore, in this study, three models of ligaments and joints in the human middle ear were employed to compare their influence on sound transmission. The three models are nonlinear with the Kelvin–Voigt and the Maxwell elements of various configurations. An exponential model of stress relaxation is proposed. The aim of this study is to show the differences in the different approaches of the human middle ear modelling. The Maxwell viscoelastic element and the modified K-V system have not yet been applied for middle ear analysis. Thus, this is a new contribution of the paper.

The paper is organized as follows: Section 2 presents the experimental setup, procedure of measurements, numerical research, and the description of mathematical models of the middle ear based on the Maxwell and the Kelvin–Voigt approach. Section 3 describes results of the numerical analysis and experimental verification of stapes vibrations as a frequency response function—The effects of external excitation, ligaments stiffness ratio, and relaxation time were investigated. Finally, a discussion of results and overall conclusions are given in Section 4 and Section 5.

## 2. Materials and Methods

Experimental and numerical approaches were employed to study the ME dynamics. First, an experiment on the human temporal bone was performed to obtain the real middle ear transfer function (frequency response function—FRF) and to evaluate parameters (stiffness and damping) for theoretical models. Next, three mathematical models of the ME were built and then the numerical procedure was developed. Finally, on the basis of the experimental outcomes, some parameters for numerical simulations were obtained. Thus, the experimental FRF is required to fit the output of numerical results with the real intact middle ear and validate the structure of the model. More detailed steps of the analysis are described in the subsequent subsections.

### 2.1. Experimental Procedure

To obtain the transfer function of the ME and to verify the middle ear model validity, experiments were performed on six specimens of the human temporal bone. A standard otolaryngology procedure was employed to prepare the specimens. The soft tissue was removed and the antromastoidectomy with posterior tympanotomy was performed. The mastoid facial nerve was removed to visualize the stapes arch and the stapes footplate. An artificial external ear canal of 25 mm in length and 9 mm in diameter was then attached to the bone with epoxy resin. The artificial canal was equipped with two ports, one for an ear microphone (ER-7C Etymotic Research) and one for a sound source (ER2 Etymotic Research) as presented in Figure 1. The artificial canal was closed with a glass plate to create a sound sealed chamber. Pieces of reflective tape (weighing less than 0.05 mg) were placed on the footplate of the stapes. The temporal bone specimen was then embedded in dental cement and put in the temporal bone holder (Storz). Measurements were performed on an anti-vibration table inside a sound booth. The same procedure was applied in other experiments, which are described in [16,21,22]. The middle ear was stimulated by a discrete chirp signal with a frequency ranging from 0.4 to 8 kHz to get the ear characteristics in the range of hearing. The stapes footplate velocity was measured with a Laser Doppler Vibrometer (LDV) system from Polytec GmbH, Waldbronn, Germany. The system consists of a Polytec controller (OFV-5000) and a laser head (OFV-534). A helium–neon laser beam was directed onto retro-reflective targets on the stapes footplate through the artificial ear canal. The measured velocity of the stapes was transferred from the controller to the National Instruments DAQ card (NI6210) and then through the PC to the DasyLab software in order to capture data. A scheme of the experimental setup is presented in Figure 1.

### 2.2. Middle Ear Models

In the human middle ear, the malleus is attached to the tympanic membrane via the handle of the malleus and to the temporal bone by means of the lateral malleal ligament, the tensor tympani tendon, and the anterior mallear ligament (Figure 2). The head of the malleus lies in the epitympanic recess, where it articulates with the next auditory ossicle, the incus, via the incudomallear joint (IMJ). The incus consists of a body and two processes. The body articulates with the malleus, the short process is attached to the posterior wall of the middle ear, and the long process joins the stapes with the incudostapedial joint (ISJ). The incus is connected to the bone with the superior malleal and the posterior incudal ligaments. The stapes, the smallest bone in the human body, joins the incus to the oval window of the inner ear. The footplate of the stapes is connected to the oval window by means of the annular ligament (AL) and with the temporal bone via the stapedius tendon. There are two muscles that serve a protective function in the middle ear: the tensor tympani and the stapedius muscle. The stapedius muscle contracts in response to a loud noise, thus inhibiting the vibrations of the ossicles, and reducing the transmission of sound to the inner ear. This effect, known as the acoustic reflex is neglected in this study when modelling the sound transmission process. Moreover, only the parts directly responsible for sound transmission in the hearing range are taken into consideration. Thus, the proposed middle ear (ME) models consist of masses connected by springs (*k*) and dashpots (*c*) that are combined in the Maxwell viscoelasticity type depending on assumed models, shown in Figure 2 and Figure 3. The ossicle are represented by three masses: the malleus ($m_{M}$), the incus ($m_{I}$), and the stapes ($m_{S}$). The lumped masses ($m_{M}$, $m_{I}$, $m_{S}$) can move horizontally on a base. This assumption eliminates a possibility of rotation but does not influence primary vibration modes. The ossicles are connected to each other by the joints IMJ and ISJ, and to the temporal bone with the ligaments AML, PIL, and AL represented by the viscoelastic model of material consisting of dashpots and springs connected in series or in parallel. In this way, the so called modified Maxwell model, also known as the Zener model, is obtained (Figure 2). This configuration provides a more realistic behaviour of the tissues than the K-V model (Figure 3), which is applied the most often in the literature. Damping and stiffness properties of the cochlea ($c_{c}$ and $k_{c}$), and of the tympanic membrane ($c_{TM}$, $k_{TM}$) are described by the K-V model both in Figure 2 and Figure 3. Moreover, the AL has nonlinear (cubic) stiffness characteristics denoted by $k_{AL3}$, as reported in [12]. The ear is stimulated by an outer signal, Qcos(ωt), acting on the malleus through the tympanic membrane. The joints and ligaments that are the most important for sound transfer are marked in bold in the model shown in Figure 2. These elements are modelled in the study.

For the model presented in Figure 2, the total stiffness (*k*) of the joints and ligaments is divided into two parts $k^{\alpha }$ and $k^{\beta }$, where $k^{\beta }$ is connected in series with a dashpot. The total stiffness (*k*) is given by equation
(1)$$k=k^{\alpha }+k^{\beta }$$

In order to regulate both parts of the stiffness coefficient α and β the following dependence is introduced
(2)$$k^{\alpha }=\alpha k,\;k^{\beta }=\beta k,\;\alpha +\beta =1$$

This also makes it possible to go to the pure K-V model by eliminating $k^{\beta }$. The middle ear with K-V viscoelasticity is presented in Figure 3.

These two models (Figure 2 and Figure 3) are analysed in terms of their behaviour and its effect on sound transfer in the ear. Additionally, in the model with the K-V viscoelasticity (Figure 3), the relaxation phenomenon is considered using a modified Young’s modulus (*E*) of the joints and ligaments having the form
(3)$$E=E_{0}+E_{1}e^{-t/{\tilde{t}}_{1}}$$
where $E_{0}$ and $E_{1}$ denote the final and the changeable component of the elastic (Young’s) modulus of the tissue. Then, the preliminary modulus is $E_{0}+E_{1}$. ${\tilde{t}}_{1}$ denotes the relaxation time; therefore, a new stiffness, $k_{r}$ is defined as
(4)$$k_{r}=k_{0}\left(1+\frac{E_{1}}{E_{0}}e^{-t/{\tilde{t}}_{1}}\right)$$
where, $k_{0}=\frac{AE_{0}}{L}$, *A* and *L* denote the cross section and the length of the joint or ligament, respectively. Consequently, three models of the ME were investigated and compared. Their description is put together in Table 1.

Governing differential equations of motion for the three models are derived, based on the Lagrange equation of the second type in the form
(5)$$\begin{array}{c} \frac{d}{dt}\left(\frac{\partial T}{\partial {\dot{x}}_{i}}\right)-\frac{\partial T}{\partial x_{i}}+\frac{\partial V}{\partial x_{i}}+\frac{\partial \Phi }{\partial {\dot{x}}_{i}}=Q_{i}. \end{array}$$

Afterwards, a potential energy(*V*) and the Rayleigh function of dissipation (Φ) are defined for each model, and the kinetic energy (*T*) is the same for all presented models, given as follows
(6)$$T=\frac{1}{2}\left(m_{M}{\dot{x}}_{M}^{2}+m_{I}{\dot{x}}_{I}^{2}+m_{S}{\dot{x}}_{S}^{2}\right)$$

The potential energy, the Rayleigh function and derivation of equations of motion are presented in appendixes. Here, only the final, dimensionless form of the differential equations of motion, that are used to build numerical models, are shown below
Model 1—Maxwell viscoelasticity
(7)$$\begin{array}{c} {\ddot{x}}_{1}+{\tilde{k}}_{11}x_{1}+{\tilde{k}}_{12}x_{2}+{\tilde{k}}_{17}y_{12}+{\tilde{c}}_{11}{\dot{x}}_{1}+{\tilde{c}}_{14}{\dot{y}}_{1}=q_{0}cos\left(\Omega \tau \right)\\ {\ddot{x}}_{2}m_{2}+{\tilde{k}}_{21}x_{1}+{\tilde{k}}_{22}x_{2}+{\tilde{k}}_{23}x_{3}+{\tilde{k}}_{28}y_{23}+{\tilde{c}}_{22}{\dot{x}}_{2}+{\tilde{c}}_{25}{\dot{y}}_{2}+{\tilde{c}}_{27}{\dot{y}}_{12}=0\\ {\ddot{x}}_{3}m_{3}+{\tilde{k}}_{32}x_{2}+{\tilde{k}}_{33}x_{3}+{\tilde{c}}_{33}{\dot{x}}_{3}+{\tilde{c}}_{36}{\dot{y}}_{3}+{\tilde{c}}_{38}{\dot{y}}_{23}+\gamma_{3}{x_{3}}^{3}=0\\ {\tilde{k}}_{44}y_{1}+{\tilde{c}}_{41}{\dot{x}}_{1}+{\tilde{c}}_{44}{\dot{y}}_{1}=0\\ {\tilde{k}}_{55}y_{2}+{\tilde{c}}_{52}{\dot{x}}_{2}+{\tilde{c}}_{55}{\dot{y}}_{2}=0\\ {\tilde{k}}_{66}y_{3}+{\tilde{c}}_{63}{\dot{x}}_{3}+{\tilde{c}}_{66}{\dot{y}}_{3}=0\\ {\tilde{k}}_{71}x_{1}+{\tilde{k}}_{77}y_{12}+{\tilde{c}}_{72}{\dot{x}}_{2}+{\tilde{c}}_{77}{\dot{y}}_{12}=0\\ {\tilde{k}}_{82}x_{2}+{\tilde{k}}_{88}y_{23}+{\tilde{c}}_{83}{\dot{x}}_{3}+{\tilde{c}}_{88}{\dot{y}}_{23}=0 \end{array}$$
where, new dimensionless coefficients of stiffness ($k_{ij}$), damping ($c_{ij}$), and coordinates ($x_{1}-x_{3}$) are defined in Appendix A—Equations (Equation 14) and (Equation 15). New stiffness and damping coefficients are a combination of the dimensional coefficients as shown in Equation (Equation 14).
Model 2—Kelvin–Voigt viscoelasticity
(8)$$\begin{array}{c} {\ddot{x}}_{1}m_{1}+k_{11}x_{M}+k_{12}x_{I}+c_{11}{\dot{x}}_{M}+c_{12}{\dot{x}}_{I}=q_{0}\cos\left(\Omega \tau \right)\\ {\ddot{x}}_{2}m_{2}+k_{21}x_{M}+k_{22}x_{I}+k_{23}x_{S}+c_{21}{\dot{x}}_{M}+c_{22}{\dot{x}}_{I}=0\\ {\ddot{x}}_{3}m_{3}+k_{32}x_{I}+k_{33}x_{S}+c_{32}{\dot{x}}_{I}+c_{33}{\dot{x}}_{S}+\gamma_{3}{x_{S}}^{3}=0 \end{array}$$
where new dimensionless coefficients $k_{ij}$, $c_{ij}$ and coordinates ($x_{1}-x_{3}$) are defined in Appendix B—Equations (Equation 19) and (Equation 20). New stiffness and damping coefficients are a combination of the dimensional coefficients as shown in Equation (Equation 19).
Model 3—Kelvin–Voigt viscoelasticity with relaxation


The differential equation of motion for Model 3 is the same as that of Model 2 (Equation (Equation 8)) but now the system stiffness is expressed using Equation (Equation 4). Thus, $k_{ij}$ in Equation (Equation 8) is replaced by $k_{ijr}$; therefore, the governing equations of motion take the form
(9)$$\begin{array}{c} {\ddot{x}}_{1}m_{1}+k_{11r}x_{M}+k_{12r}x_{I}+c_{11}{\dot{x}}_{M}+c_{12}{\dot{x}}_{I}=q_{0}\cos\left(\Omega \tau \right)\\ {\ddot{x}}_{2}m_{2}+k_{21r}x_{M}+k_{22r}x_{I}+k_{23r}x_{S}+c_{21}{\dot{x}}_{M}+c_{22}{\dot{x}}_{I}=0\\ {\ddot{x}}_{3}m_{3}+k_{32r}x_{I}+k_{33r}x_{S}+c_{32}{\dot{x}}_{I}+c_{33}{\dot{x}}_{S}+\gamma_{3}{x_{S}}^{3}=0 \end{array}$$

Taking into account the transformation presented in Equation (Equation 20), the dimensionless stiffness is defined
(10)$$\begin{array}{c} k_{ijr}=k_{ij}\left(1+\frac{E_{1}}{E_{0}}e^{-\tau /t_{1}}\right) \end{array}$$

### 2.3. Numerical Procedure

On the basis of the mathematical models, presented in Section 2.2, numerical models are built in MATLAB Simulink. Next, numerical calculations are preformed using the Runge–Kutta 4th order method (ode45) with a variable step size and a relative tolerance of $10^{-10}$. In the numerical experiments, the parameters of the model presented in Table 2 are used. The parameters of stiffness (*k*), damping (*c*), and mass (*m*) are taken from [14]. The rest of them, which are new in the presented models, are estimated based on experimental findings in such a way that the numerical outcomes (FRF) correspond to the experimental ones, mainly the frequencies and amplitudes of resonances have been compared.

At the beginning, in Model 1 the stiffness ratio α is assumed to be 0.96, because some researchers claim that the ligaments and tendons should be treated as collagen fibres and the surrounding proteoglycan-rich matrix [10,23,24,25]. Thus, the mechanics of these tissues is also modelled using the Maxwell-type viscoelastic material, where the elastic modulus of the matrix and the fibre is $E_{m}$ = 130.4 MPa and $E_{f}$ = 3500 MPa, respectively. The total modulus $E=E_{m}+E_{f}$. The coefficient α (according to Equations (Equation 1) and (Equation 2)) represents the relationship between the total and the matrix Young modulus in the form $\alpha =E_{m}/E$.

## 3. Results

A frequency response function (FRF) of the middle ear, which is sometimes also called a transfer function, is most commonly used by ear researchers for both engineering and medical purposes. The FRF is a mathematical representation of the relationship between the input and the output of a system. Usually, the pressure applied to the ear is the input, whereas the stapes vibration velocity caused by the pressure is the output. Here, the FRF obtained from the experiment is compared with the numerical results yielded for the three models. The analysis focuses on the stapes vibration because of its importance for sound transmission to the inner ear.

### 3.1. Frequency Response Function

Stapes vibrations measured in the experiment and presented in Figure 4 are normalized to the pressure of 0.55 Pa in order to compare the results with the numerical outcomes. Then, the FRF is unified and does not depend on the excitation level. The mean value of the stapes velocity from the experiment is marked with a blue line in Figure 4. Additionally, the results of numerical simulations (black, green, and red lines) are shown together to compare and verify three models with experimentally obtained outcomes.

Experimental results show that the first resonance of the intact middle ear is about 1 kHz and the second one is about 5.5 kHz. Similar resonances are expected for the analysed models; therefore, the parameters of Model 1 (Maxwell) as well as Models 2 and 3 (K-V) given in Table 2 differ with respect to the damping coefficient. Damping coefficients in the models are adopted so that the velocity amplitude would be the same as in the experiment. The experimental and numerical resonances obtained from the Models 1 and 2 are convergent, that is expected in case of small β, used in calculations, whereas for Model 3 a shift of the second resonance is observed. The resonance offset occurs in spite of the fact that stiffness coefficients are the same and it is caused by the relaxation effect. It should be remembered that a pure harmonic stimulation leads to harmonic vibration of the ossicles in the intact human ear. This may, however, be changed by altering the model parameters or excitation amplitude; therefore, the problem of stapes vibration is analysed with special care in subsequent sections, where several key parameters are analysed.

### 3.2. Influence of Excitation Amplitude

The excitation amplitude is a key parameter influencing the stapes dynamics. There is no doubt that a higher excitation amplitude causes stronger vibrations of the stapes, as presented in Figure 5; however, a very strong excitation may cause irregular motion, especially in nonlinear systems. In the case under analysis, the nonlinearity of the annular ligament ($k_{AL3}$) causes an unexpected behaviour of the stapes shown in Figure 6. The vertical dashed line marks the border where pure harmonic vibrations go to periodic oscillations. The periodic vibrations consist of two harmonic answers with different amplitudes but of the same frequency. This effect can be called a disturbed harmonic response (DHR). The blue line shows the start point of the first disturbance in harmonic response and the green one shows the higher order disturbances. That means the output signal ($x_{3}$) reveals more than one additional harmonic. The DHR starts earlier for Model 1 than for Models 2 and 3. The change of motion is observed only for the stapes, while the malleus and the incus vibrate harmonically all time. This behaviour is presented in Figure 7 as phase diagrams (black) with Poincaré points (red) near the first resonance.

The phase diagram, only for Model 1, are shown in Figure 7, because Models 2 and 3 demonstrate the same behaviour. The stapes motion depends on the excitation amplitude but the period of vibrations is still the same and equal to 1T. 1T means that the response vibrations have period 1 in relation to the excitation frequency (the period of the response is exactly the same as the excitation period).

### 3.3. Effect of Stiffness Ratio

In the FRF shown in Figure 4, α=0.96. Now the influence of α (also β, see Equation (Equation 2)) on dimensionless amplitude of the stapes vibration ($a_{3}$) is analysed in Figure 8. An increase in α (the spring parallel to the pure Maxwell viscoelastic element) causes a nonlinear increase in the vibrations amplitude. This is especially visible at higher values of the external excitation (5Q and 10Q). Thus, the stiffness ratio does not change the type of motion, but has an effect on the stapes vibration amplitude.

### 3.4. Effect of Relaxation Time

Based on Model 3, the dimensionless relaxation time ($t_{1}$) is analysed in terms of its influence on the stapes vibrations. The relaxation effect causes a shift in the second resonance, which can be observed in Figure 4. The second resonance ($\Omega_{2}$) shift increases with $t_{1}$, as shown in Figure 9 while the vibrations dimensionless amplitude ($a_{3}$) decreases.

The change in the frequency ($\Omega_{2}$) and the amplitude ($a_{3}$) is nonlinear, as shown in Figure 10. When $t_{1}>3000$, the effect of relaxation does not affect the stapes motion; however, when the relaxation time is short, this effect cannot be neglected. Thus, to model the middle ear properly, the relaxation phenomenon should be taken into consideration. Interestingly, the relaxation time does not influence the first resonance. This is probably because the stapes is fixed to the cochlea by means of the AL, which is modelled as nonlinear. In nonlinear systems, their behaviour is not deterministic, which means the system answer is not proportional to input parameters (nonlinear phenomena are observed).

## 4. Discussion

The presented lumped masses model has some simplifications during modelling, e.g., a one directional horizontal motion of the ossicles that do not influence the primary vibration modes. The first and the second vibration modes of the stapes are longitudinal vibrations, usually analysed in the hearing frequency range [3,4]. The middle ear dynamics are rarely analysed in the literature in terms of the effect of joints and ligaments. Investigators usually assume that the human tissue exhibits viscoelastic properties of the K-V type [4,5,6]. Sometimes, only FEM models are built to analyse the response of ossicles under different conditions [26,27,28,29,30]; however, these models presented in the literature do not show the effect of relaxation that is typical for the human tissue, as reported in [7,9,10,11] and therefore should be taken into consideration. Thus, our study is the first attempt to compare viscoelastic properties of ligaments and joints in the middle ear on the stapes behaviour. The three models analysed here have the same first resonance as that obtained from the experiment performed by the authors of [14,16] and reported in other papers [5,31]; however, the shift of the second resonance, due to the relaxation effect in Model 3, is considered by us to be a new and interesting phenomenon.

The stapes vibration amplitudes do not differ from each other in the three models, but strong excitation causes DHR, which is earlier observed in Model 1 than in Models 2 and 3. Interestingly, the relaxation time in Model 3 is of key importance only for the second resonance. The relaxation time increases the resonance frequency and decreases the stapes amplitude. The significance of the relaxation time was also reported in [10,11], but not for the middle ear dynamics. Usually, the relaxation time of tissues is long, about 100–500 s [32,33,34]; therefore, in the real conditions when the sound approaching the ear is dynamic, the shift of the second resonance cannot be observed in the middle ear systems. The presented model of relaxation can be used in future to model new bio-materials and implants.

The Maxwell and Kelvin–Voigt models without relaxation are suitable for analysing rapidly changing phenomena, when the time of exposure is short and then the relaxation effect may be neglected; however, it is recommended to use the model with relaxation when the exposure time is long, for example, to observe movements of the ossicular chain during changes in static pressure [35].

In the future, the problem of middle ear nonlinearity in both the Maxwell and the K-V models with relaxation should be investigated.

## 5. Conclusions

The middle ear dynamics is sensitive for many different conditions and parameters; therefore, the proper modelling is of special importance. Here, the three models of different structures are compared. The most important conclusions that can be drawn from this study are as follows:two main resonances of the middle ear are observed in the experiment on the human temporal bones;experimental tests, performed to track the parameters and to test the mathematical models outputs, prove that the models give consistent results with experimental outcomes for the tested preliminary parameters;the middle ear models with the Maxwell and the K-V viscoelasticity yield very similar results, particularly when the relaxation time is short and the excitation amplitude is small;a longer relaxation time causes an offset of the second resonance in the human middle ear;the effect of disturbed harmonic response (DHR) occurs at slightly different values of external excitation in the model with the Maxwell viscoelasticity, when compared to the Kelvin–Voigt model;the analysed ME model of the Maxwell type of viscoelasticity is sensitive for the stiffness ratio (α) which changes the value of the resonance amplitude.

The study brings some important aspects in modelling of the middle ear and can be used in practice to develop both passive and active middle ear implants, made of new bio-materials. On the whole, the active one should be controlled in the proper manner to avoid disturbances in harmonic responses. The proposed ME models can be applied for prediction of the middle ear behaviour and for control purposes.

## Figures and Tables

**Figure 1 materials-13-03779-f001:**
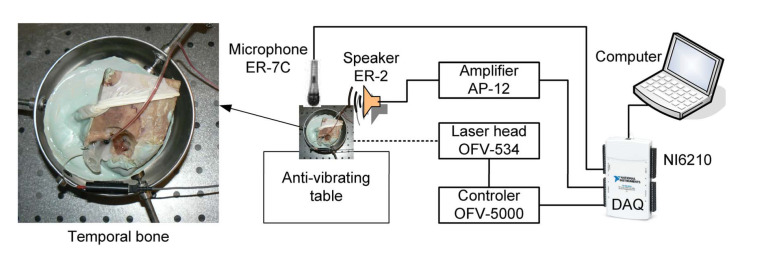
Experimental setup with a Laser Doppler Vibrometer (LDV) system for measuring stapes vibration.

**Figure 2 materials-13-03779-f002:**
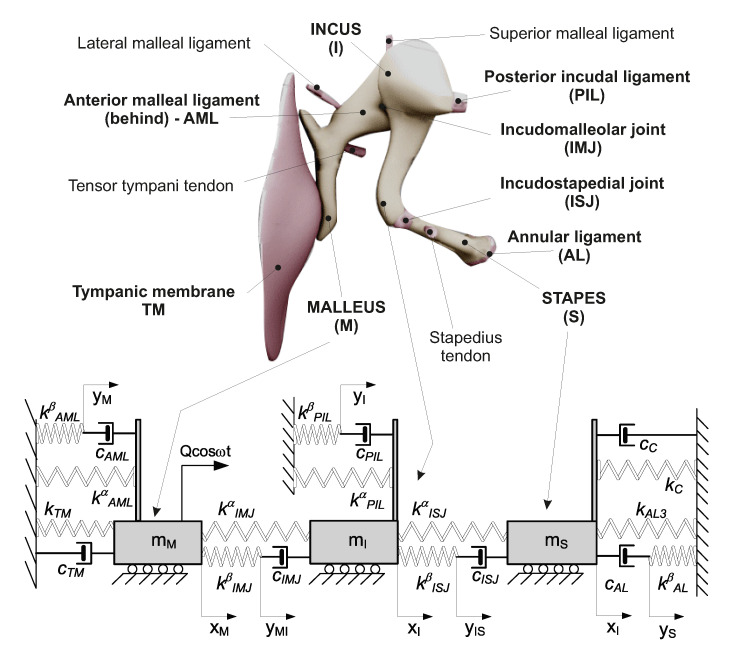
Schematic representation of the human middle ear and a three degrees of freedom model of the human middle ear with viscoelasticity of the Maxwell type (Model 1).

**Figure 3 materials-13-03779-f003:**
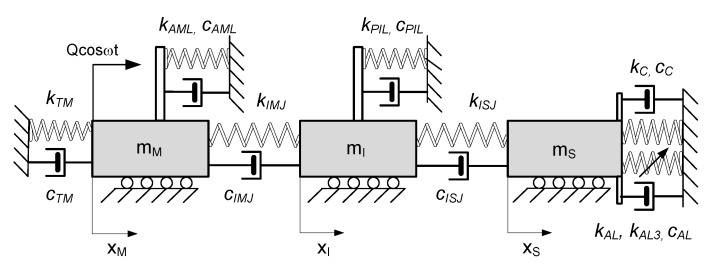
Three degrees of freedom model of the human middle ear with the Kelvin–Voigt type of viscoelasticity used for analysing systems with and without relaxation effect (Models 2 and 3).

**Figure 4 materials-13-03779-f004:**
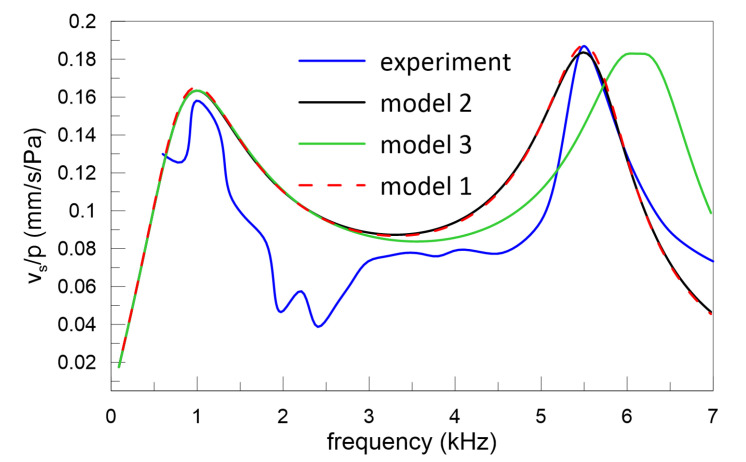
Transfer function of the middle ear from experiment (blue line) and numerical simulations (black, green, and red lines) expressed as the ratio of stapes velocity to sound pressure ($v_{s}/p$) versus excitation frequency.

**Figure 5 materials-13-03779-f005:**
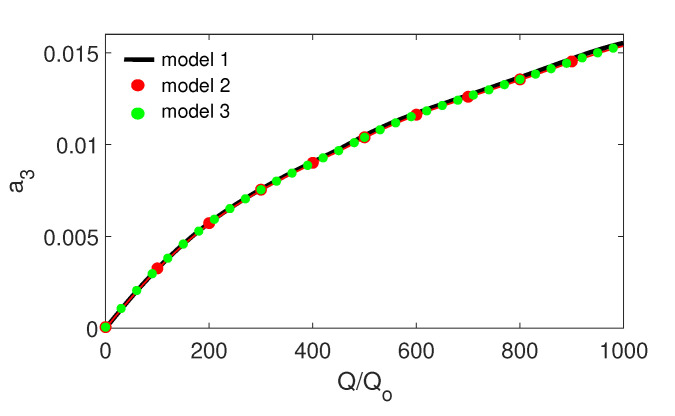
Influence of the external excitation amplitude (*Q*) on the stapes dimensionless amplitude ($a_{3}$) near the first resonance.

**Figure 6 materials-13-03779-f006:**
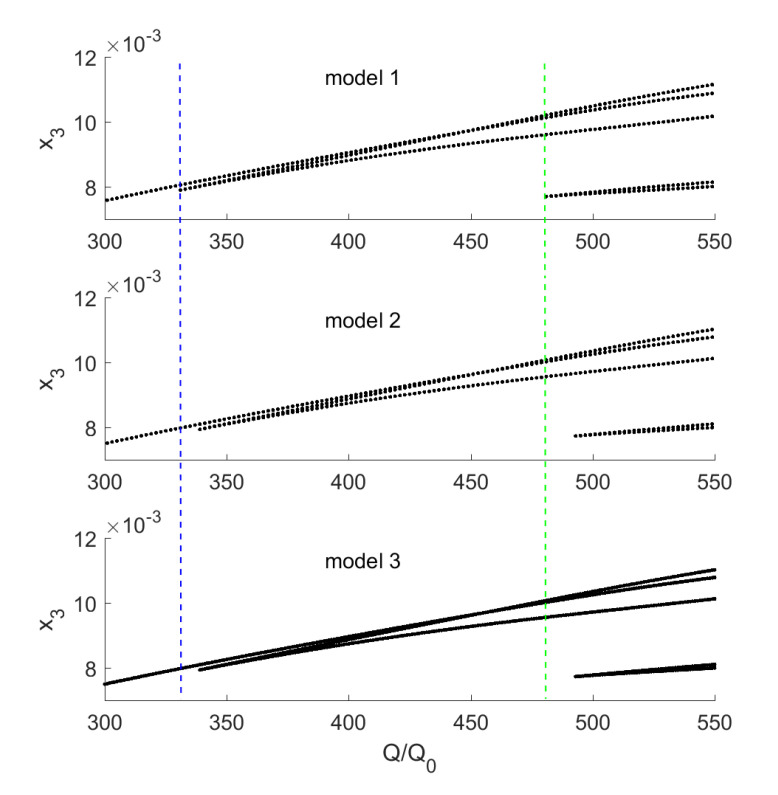
Influence of the external excitation amplitude (*Q*) on the stapes motion ($x_{3}$) near the first resonance.

**Figure 7 materials-13-03779-f007:**
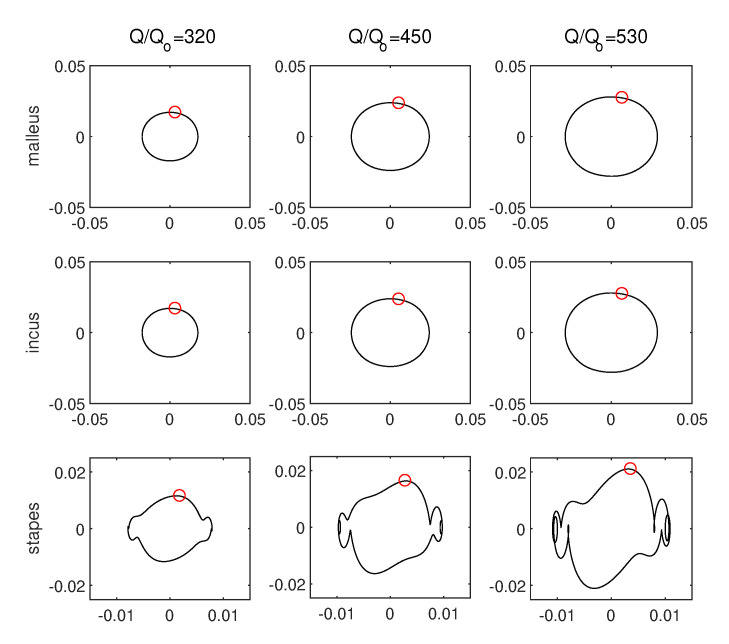
Ossicles motion presented as phase diagrams (black) with Poincaré points (red) for Model 1 and different external excitation amplitudes ($Q/Q_{o}$).

**Figure 8 materials-13-03779-f008:**
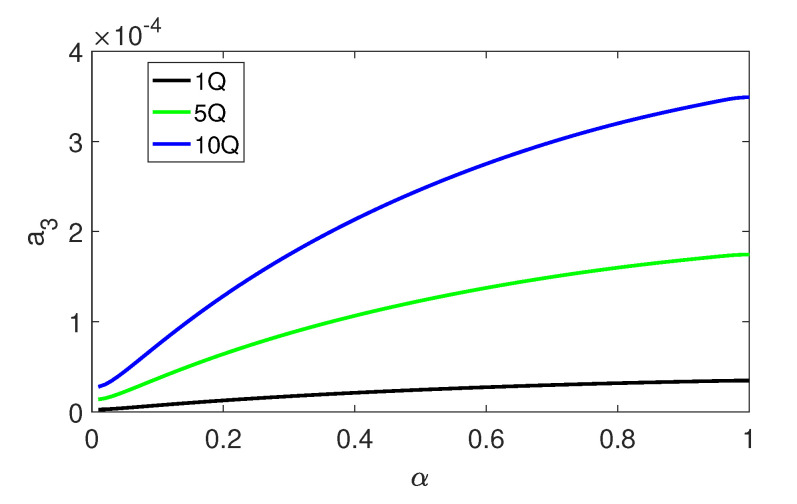
Influence of the stiffness ratio (α) in Model 1 on the stapes dimensionless amplitude ($a_{3}$).

**Figure 9 materials-13-03779-f009:**
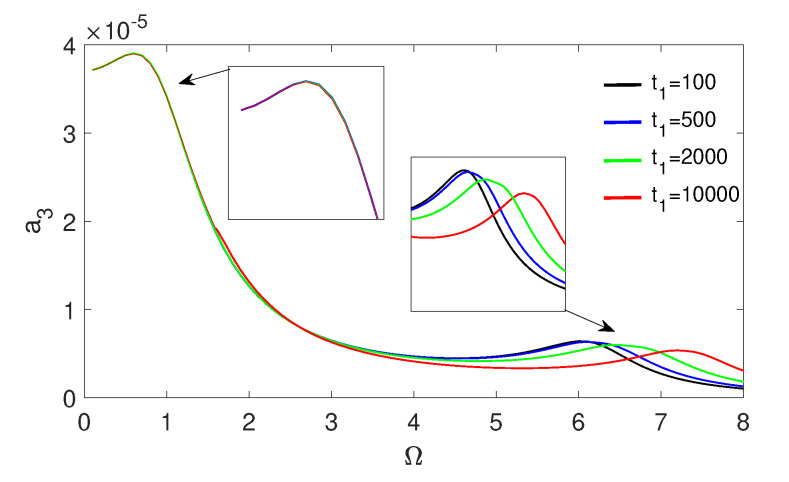
Resonance curves of the stapes ($a_{3}$) in Model 3 for different relaxation times ($t_{1}$).

**Figure 10 materials-13-03779-f010:**
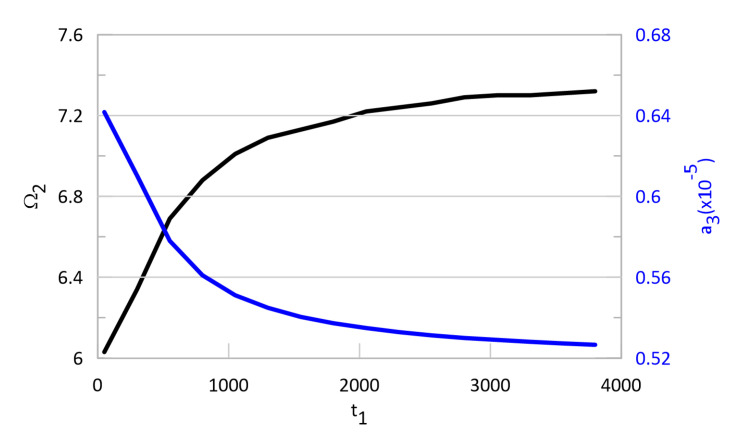
Second resonance dimensionless frequency $\Omega_{2}$ (black curve) and dimensionless amplitude $a_{3}$ (blue curve) as a function of relaxation time $t_{1}$. Dimensionless frequency $\Omega_{2}$ is related to $\omega_{0}$.

**Table 1 materials-13-03779-t001:** Characteristics of three degrees of freedom (dof) models of the middle ear.

Model 1	Model 2	Model 3
Maxwell viscoelasticity	Kelvin–Voigt viscoelasticity	Kelvin–Voigt viscoelasticity with relaxation effect
Figure 2	Figure 3	Figure 3
Equations (Equation 7) and (Equation 15)	Equations (Equation 8) and (Equation 20)	Equations (Equation 9) and (Equation 10)

**Table 2 materials-13-03779-t002:** Parameters of the middle ear used in numerical simulations, taken from experimental validation.

Stiffness	Damping	Damping	Other
(Model 1,2,3)	Maxwell (Model 1)	Kelvin–Voigt (Model 2,3)	
kTM=0.3 mN/μm	cTM = 378 mNs/mm	cTM = 60 mNs/mm	mM = 25 mg
kAML=0.8 mN/μm	cAML = 0.4 mNs/mm	cAML = 275 mNs/mm	mI = 28 mg
kIMJ=1000 mN/μm	cIMJ = 359 mNs/mm	cIMJ = 359 mNs/mm	mS = 1.78 mg
kPIL=0.4 mN/μm	cPIL = 55 mNs/mm	cPIL = 55 mNs/mm	Q0=1.2e−4 N
kISJ=1.35 mN/μm	cISJ = 7.9 mNs/mm	cISJ = 7.9 mNs/mm	α=0.96
kAL=0.623 mN/μm	cAL = 4000 mNs/mm	cAL = 2 mNs/mm	E1/E0=0.5
kC=0.2 mN/μm	cC = 11 mNs/mm	cC = 1.7 mNs/mm	t1=0.0884
km=0.85 mN/μm			
kAL3=13 mN/μm3

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
