# Peer review of "Biomechanics of the Human Middle Ear with Viscoelasticity of the Maxwell and the Kelvin–Voigt Type and Relaxation Effect"

_materials, 2020, doi:10.3390/ma13173779_

Round 1
Reviewer 1 Report
The authors described three models to study the vibrations of the human ear. They compared a Maxwell model, a Kelvin-Voigt model, and a modified Kelvin-Voigt (based on the addition of time relaxation). The mathematical description is detailed, however, I am wondering what are the novel contributions of your results and the potential applications. I recommend stressing those points. I also include some comments divided into Major and Minor:
Major
- I miss an introductory sentence, and a sentence describing your conclusions. I also miss a brief comment about potential applications.
- I recommend doing a deep revision of the use of English.
- Overall, I would focus the state-of-the-art of the Introduction more in line with your study. It should be reduced, highlighting only what is relevant for your article.
- You mention that the head of the stapes “articulates”, and in Figure 1, the bones seem to allow rotations at their joints. Why you did not consider them? If the bones can rotate, I think that you cannot neglect this effect.
- Why does this provide more realistic behavior?
- Are those the only differences? According to the figures, there are more.
- Before equation (4). If I understood correctly, if t approaches infinite, E1 is not the final elastic modulus.
- Eq (7). Why the last term is at the 4th power?
- I would include the mathematical expressions in an appendix, I think that their contribution is minimal to the whole text. I would avoid repetitions of the mathematical expressions when it is possible.
- Please, describe better figure 4 both in the text and in the figure caption.
- Table 2. So, did you obtain these values after fitting the model with experimental data? If so, it is not completely clear in the text.
- How did you obtain the polyharmonic motion from Figure 6?
- What did you use alfa=0.96? what is the relation with the fact of considering the collagen fibers?
- What the first resonance does not experience any difference? Please, discuss it.
- It seems weird to mention all conclusions as bullet points.
- It was expected to me, it is not surprising.
- What is the take-home message of this study? What is the study important and novel?
Minor
- L26 and L27. “However. Nevertheless”, there is a repetition, but above of all, neither “However” nor “Nevertheless” seem to fit here.
- L29 to 31. This sentence does not make sense. Ligament models are linear? Why the word “whereas”?
- L39 and 40. Neither “Thus” not “However” seem to fit. The first one because there is no cause-effect, and the second because there is no contrast between sentences.
- In lines 29 and 30, only Kelvin-Voigt models are introduced, whereas in lines 43 and 44, Kelvin-Voigt and Maxwell. Why?
- A comma is missing between “incus” and “the superior”?
- “investigated”
- As “the so called the modified Maxwell model”, do you mean that you define with this name, or that it is known in the literature with this name?
- Include reference for Zener model.
- Are supraindexes a and b in Figure 1 the same as alfa and beta in the text? If so, please, unify them.
- Before equation (4). Why “therefore”?
- Why “Consequently”?
- Before expression (11). This is confusing, tau seems not introduced in expression (11). Why has omega been changed from lower-case to upper-case? Why the change of the coordinates?
- “Ear researchers” sounds weird.
- Remove “is” after “system”.
- Include reference to DasyLab.
- What do you mean with “the results are unified”? Do you show the mean values?
- Figures 5 to 10, units are missing.
- What is “1T”?
- Figure 7. What are the horizontal and vertical axes? Position and velocity?
- You mention that it does not affect the dynamics, but affect the vibration amplitude. So, what do you consider by the dynamics? Only velocity?
- “Interestingly”.
- L229-230. The verb is missing.
Author Response
Response to the review 1
Authors would like to thank the reviewer very much for the important feedback that helped to improve the quality of the paper. All changes, suggested by the reviewer, are marked in red colour in the manuscript.
Major remarks:
Remark 1:
I miss an introductory sentence, and a sentence describing your conclusions. I also miss a brief comment about potential applications.
Answer:
Thank you for your suggestion. The missing sentences are written in Introduction, that is completely rebuilt and also Section Discussion and Conclusion is completed
Remark 2:
I recommend doing a deep revision of the use of English.
Answer:
English has been improved in the manuscript
Remark 3:
Overall, I would focus the state-of-the-art of the Introduction more in line with your study. It should be reduced, highlighting only what is relevant for your article.
Answer:
Thank you for your suggestion. Introduction is completely changed. Now, it is shorter and focused exactly on the topic of our study i.e. middle ear model, relaxation of ligaments and the problem of nonlinearity.
Remark 4:
You mention that the head of the stapes “articulates”, and in Figure 1, the bones seem to allow rotations at their joints. Why you did not consider them? If the bones can rotate, I think that you cannot neglect this effect.
Answer:
My sentence was mistakenly written. The ossicles can rotate only if they are connected (each other) by joints. However, a base of the stapes (the head of the stapes is introduced instead of the head) is connected to the oval window by means of the ligament (AL). It has been improved.
Remark 5:
Why does this provide more realistic behavior.
Answer:
The modified Maxwell model (Zener model, mentioned in line 123 – old version) is more realistic because a spring and a dashpot are connected each other in series and additionally with the second spring in parallel. Therefore, the Zener model has advantages of both the Maxwell and the K-V structure. It can model both the creep and relaxation effect simultaneously. The appropriate comment is given in the manuscript.
Remark 6:
Are those the only differences? According to the figures, there are more.
Answer:
Now, it is discussed in the manuscript
Remark 7:
Before equation (4). If I understood correctly, if t approaches infinite, E1 is not the final elastic modulus.
Answer:
You are right E0 is final elastic modulus. E1 can be named as preliminary one. It is corrected. Thank you.
Remark 8:
Eq (7). Why the last term is at the 4th power?
Answer:
In Eq. 7 the last term is at the 4th power because stiffness of the AL is assume as a cubic. Then, (kAL3xs3) is spring force. The relation between a spring force (F) and a potential energy (V) is F=dV/dx (or F=-dV/dx, depends on coordinate system). Thus the power is 4. The comment about cubic nonlinearity is added in Section 2.2 (old Section 2).
Remark 9:
I would include the mathematical expressions in an appendix, I think that their contribution is minimal to the whole text. I would avoid repetitions of the mathematical expressions when it is possible
Answer:
Some of the mathematical expressions are included in an appendix. Moreover. According to the second reviewer comment, structure of the paper is changed a bit.
Remark 10:.
Please, describe better figure 4 both in the text and in the figure caption
Answer:
It is done.
Remark 11:
Table 2. So, did you obtain these values after fitting the model with experimental data? If so, it is not completely clear in the text.
Answer:
Yes, we have obtained these values after fitting the model with experimental data. The comment is put in Section 2 (old Section 3).
Remark 12:
How did you obtain the polyharmonic motion from Figure 6?.
Answer:
The points in Fig.6 (that create lines) are collected from the phase space..(x-v) when the trajectory cross the horizontal axis (velocity v=0). We do not know anything about a vibration period but it means the harmonic answer is perturbed in relation to pure harmonic signal. The answer is regular, but has a second disturbed component. The additional analysis, shown in Fig. 7, let us claim that the answer is periodic - deformed harmonic. Since the output vibrations have the period 1T – related to the external excitation period (the same as excitation) we decided to call them as polyharmonic. Indeed, it could be mistaken. Now, it is corrected in the manuscript, the motion is called as deformed harmonic response (DHR).
Remark 13:
What did you use alfa=0.96? what is the relation with the fact of considering the collagen fibers?.
Answer:
The detailed description of ligaments and tendons modelling is given in [17, 35-37 old version of the paper] The answer to the question is written in Section 2, alpha=0.96 because E=Em+Ef, then alfa=Em/E. However, the change of alpha is also analysed in the paper (Section 4.2)
Remark 14:
What the first resonance does not experience any difference? Please, discuss it.
Answer:
We think, you mean Model 3 because for example in Model 1 there are some differences e.g. in the stapes amplitude. Indeed, in Model 3 the relaxation time (Fig.9) does not influence the first resonance. This is probably because the stapes is fixed to the cochlea by means of the AL which is nonlinear. In nonlinear systems a behaviour is not deterministic and also they exhibit unexpected phenomena. The proper comment is written in Section 3.
Remark 15:
It seems weird to mention all conclusions as bullet points.
Answer:
Now, it is modified.
Remark 16:
It was expected to me, it is not surprising.
Answer:
Now, it is modified.
Remark 17:
What is the take-home message of this study? What is the study important and novel?.
Answer:
A few words about novelty is written in Introduction (). An importance of the problem and future plans are written in Conclusions.
Minor remarks:
All the mentioned minor remarks are taken under consideration and the manuscript is improved:
- L26 and L27. “However. Nevertheless”, there is a repetition, but above of all, neither “However” nor “Nevertheless” seem to fit here.
- L29 to 31. This sentence does not make sense. Ligament models are linear? Why the word “whereas”?
- L39 and 40. Neither “Thus” not “However” seem to fit. The first one because there is no cause-effect, and the second because there is no contrast between sentences.
- In lines 29 and 30, only Kelvin-Voigt models are introduced, whereas in lines 43 and 44, Kelvin-Voigt and Maxwell. Why?
- A comma is missing between “incus” and “the superior”?
- “investigated”
- As “the so called the modified Maxwell model”, do you mean that you define with this name, or that it is known in the literature with this name?
- Include reference for Zener model.
- Are supraindexes a and b in Figure 1 the same as alfa and beta in the text? If so, please, unify them.
- Before equation (4). Why “therefore”?
- Why “Consequently”?
- Before expression (11). This is confusing, tau seems not introduced in expression (11). Why has omega been changed from lower-case to upper-case? Why the change of the coordinates?
- “Ear researchers” sounds weird.
- Remove “is” after “system”.
- Include reference to DasyLab.
- What do you mean with “the results are unified”? Do you show the mean values?
- Figures 5 to 10, units are missing.
- What is “1T”?
- Figure 7. What are the horizontal and vertical axes? Position and velocity?
- You mention that it does not affect the dynamics, but affect the vibration amplitude. So, what do you consider by the dynamics? Only velocity?
- “Interestingly”.
- L229-230. The verb is missing.
Thank you very much for valuable remarks.
Yours faithfully,
Rafal Rusinek and co-authors
Reviewer 2 Report
Reviewer comments for materials-886425 “Biomechanics of the human middle ear with viscoelasticity of the Maxwell and the Kelvin-Voigt type and relaxation effect.”
- The overall structure of the paper doesn't not lend itself to a clear presentation. Why not: Introduction, Materials and Methods, Results, Discussion?
- Passive voice is used extensively. It would be more clear and concise to use active voice.
- The aim of the paper in the abstract is not the same as that on page 12. Please clarify.
- The Instruction should use shorter paragraphs with clear, concise content. It’s very difficult to follow the logic flow.
- Line 65: “As a result, the constitutive model…” does not flow logically from the statements before.
- The introduction is overall too long. And the first two paragraphs of section 2 are basically introduction.
- The model in Figure. 1 has a lot of parameters. Are there problems with uniqueness during the fitting.
- Line 141, this is the first time that stapes is in the text. Maybe this could be emphasized in Figure 1 (top)?
- Figure 7. All the plots look the same. Could these be superimposed and color-coded in one larger plot?
- Why not fit model 3 to reproduce the second peak?
- The responses of the models don’t seem to vary much. Is that the main conclusion of the paper?
Author Response
Response to the review 2
Authors would like to thank the reviewer very much for the important feedback that helped to improve the quality of the paper. All changes, suggested by the reviewer, are marked in red colour in the manuscript.
Remark 1:
The overall structure of the paper doesn't not lend itself to a clear presentation. Why not: Introduction, Materials and Methods, Results, Discussion?.
Answer:
I was wondering about the structure and was against, because the structure of our research does not fall into a classic structure of material testing. However finally I have changed it according to your suggestion. I must claim that the paper looks better. Thank you for your suggestion.
Remark 2:
Passive voice is used extensively. It would be more clear and concise to use active voice.
Answer:
Looking at others scientific papers, also published in Materials the passive voice is preferred. Moreover, in MDPI style guide for authors (https://www.mdpi.com/authors/layout#_bookmark20 Section 4) passive voice is recommended. Therefore, we stay with the passive voice but English gramma is sometimes corrected.
Remark 3:
The aim of the paper in the abstract is not the same as that on page 12. Please clarify.
Answer:
Now, it is improved in new document.
Remark 4:
The Instruction should use shorter paragraphs with clear, concise content. It’s very difficult to follow the logic flow.
Answer:
The structure of the document has been changed after Remark 1. I believe that the logic flow is much better now.
Remark 5:
Line 65: “As a result, the constitutive model…” does not flow logically from the statements before.
Answer:
The Introduction has been shortened, rewritten and improved to avoid logical errors.
Remark 6:
The introduction is overall too long. And the first two paragraphs of section 2 are basically introduction.
Answer:
Yes, as I have mentioned before, now it is shortened and improved.
Remark 7:
The model in Figure. 1 has a lot of parameters. Are there problems with uniqueness during the fitting.
Answer:
The model has a lot of parameters because of complicated nature of the middle ear structure. Sometimes, parameters differ in the models (e.g. damping) in order to fit the numerical and experimental responses. In other words, to get the middle ear characteristic more realistic. The comment is put to Section 2.
Remark 8:
Line 141, this is the first time that stapes is in the text. Maybe this could be emphasized in Figure 1 (top)?
Answer:
“the stapes” was used the first time in line 24, however in Section 2 was used at the beginning when the human middle ear model was described, also in Fig.1. Now, after revision of Section 1 and 2, clarity of the manuscript should be better.
Remark 9:
Figure 7. All the plots look the same. Could these be superimposed and color-coded in one larger plot?
Answer:
Indeed It looked strange a bit. Now the figure is reduced to Model 1 only, and comment about the same results from Model 2 and 3 is added.
Remark 10:.
Why not fit model 3 to reproduce the second peak?
Answer:
Generally, the Model 3 fits to the experiment in the second peak in case of small relaxation time. However, we have tested different value of the relaxation time and found this phenomenon as interesting and untypical. The proper comment is done in the text.
Remark 11:
The responses of the models don’t seem to vary much. Is that the main conclusion of the paper?
Answer:
Conclusions are revised to improve clarity. Indeed, the responses are almost the same but it is not the main conclusion. We focus also on differences in case of stronger excitation and longer relaxation time.
Thank you very much for valuable remarks.
Yours faithfully,
authors
Reviewer 3 Report
Article : Materials-886425
Title : Biomechanics of the human middle ear with viscoelasticity of the Maxwell and the Kelvin-Voigt type and relaxation effect
Authors : Rafal Rusinek, Marcin Szymanski, and Robert Zablotni
Comments
The manuscript deals with a theoretical modelling and an experimental investigation about the viscoelastic properties of the human middle ear.
The introduction section clearly summarises the studies reported in the literature and the present state-of-the-art. The aim of the article is explicitly declared.
The theoretical aspect of the manuscript involves three different 3-degrees of freedom models, specifically the Maxwell model, the Kelvin-Voigt model and the Kelvin-Voigt with relaxation model. The reported equations are developed by a detailed analysis based on fundamental theory.
The measurements of the vibrations of the stapes bone are performed by a well-established technique, that grants reliability of the observed values.
The presented models satisfactorily fit the observed experimental results, as the authors point out in the discussion paragraph.
The noticeable item, that is the force of the article, is the specification of the distinct effects observed in the measurement operation, namely the influence of the excitation amplitude, the effect of the stiffness ratio and the effect of the relaxation time. Essentially the large amount of the reported experimental results gives an interesting overview of the vibration behaviour.
Certainly this article will generate a large interest in the scientific medical community, involved in the studies of the mechanical properties of the human body.
The sentences in the conclusions are supported by the experimental findings.
The manuscript appears well partitioned into properly-titled paragraphs and sub-paragraphs, following a logical sequence. The readability of the text is excellent. The quality of the figure is good. The references are updated.
I suggest the publication of the manuscript in the present form .
Author Response
Authors would like to thank the reviewer very much for a very nice and precise review. Thank you very much for your work and valuable remarks and comments.
Yours faithfully,
Authors
Round 2
Reviewer 1 Report
I appreciate the effort of the authors modifying some sections of the manuscript. However, there are still some sections that need further improvement.
In the following, I mention some comments divided into major and minor:
Major:
- In the abstract, I still miss an introduction to contextualize your study.
- Do you take into account the hysteresis of the materials? According to the mathematical expressions, it does not seem so. However, the introduction of hysteresis in the Introduction can confuse the reader.
- L64 to L71. When reading the Introduction, it seems that you will evaluate stress and strains, you will model the sound transmission, you will provide a dynamic analysis (not only kinematics), and that the model will contain all elements mentioned in the introduction. However, it does not seem so. I think that the Introduction could still be more in line with your study.
- The use of English should be revised deeply. There are still many issues. See the minor comments.
- What “characteristic”? What is the goal of doing the experiments? I think that it could be explained more clearly. I think that it is still not clear for the reader that the values in Table 2 are obtained from experiments.
- Figure 1. The picture containing the bone is not clear.
- L114-115. I still read that the stapes articulates with the oval window, but in the model there are no rotations.
- The question “Are supraindexes a and b in Figure 1 the same as alfa and beta in the text? If so, please, unify them” was not answered and not clarified.
- Expression 3. The authors mention: “E0 and E1 denote the final and the preliminary elastic modulus”, however, when t=0, E=E0+E1, not E0.
- I appreciate that the authors move some expressions to the appendices. However, the reader cannot understand the meaning of variables k11, k12, c11, c14, etc., without further explanation in the text. These parameters do not appear in Figures 2 and 3. The same for the appendices. It is almost impossible that the reader understands the expressions. If the reader cannot follow them, either they should be better introduced or removed.
- So, beta is 0.04, right? It is very low, so then it is not surprising that models 1 and 2 are almost identical in Figure 4. Are the differences between models 1 and 2 only on the beta values? I would clarify what models are 1, 2 and 3.
- Figure 6. I think that the meaning of the lines needs further description, all three black lines, blue, and green lines. In the answer to the reviewers, the authors mentioned that “polyharmonic” term was not used anymore if I am correct, but it still appears in the figure caption.
- If I understood correctly, the relaxation effect is introduced as a novel contribution of this study. However, as shown in Figure 4 and explained in section 3.4, the results are not realistic. Please, discuss this issue, since it seems very fundamental for your study.
- Why unexpected?
- L224-225. I would clarify if your model is realistic or not, and how you validate your results.
- How can you validate the results obtained in Figure 9 and 10?
- I miss some conclusions in line with your experimental data. I think that the conclusions could overall still be improved.
Minor:
- L25: “…an axis that goes…”?
- This sentence could be improved.
- L32-33. Why “however”? In most cases, the models are focused on the kinematics on the intact middle ear. What is the relation with the models of Feng and Gan? Both could analyze dynamic aspects.
- This sentence needs to be rewritten.
- “one” should be removed.
- Last “e” should be “a”?
- Should “is consisted” be replaced by “consists”?
- L128-129. “which is applied the most often in the literature” seems redundant with “widely used” already mentioned.
- “The total modulus is…”
- By “related”, do you mean “normalized”?
- “The mean value…”
- Figure 4. The number of ticks in the vertical axis is normally lower in a journal paper.
- I think “go to periodic one” should be rewritten.
- I think some words are missing.
- “also” should be just before the verb.
- This sentence should be rewritten.
- Figure 8. Is the amplitude a3 dimensionless?
- “models are suitable”?
- Figures 9 and 10 should appear before to my understanding.
- In Figure 10, f2 is omega2? Please, correct it.
Author Response
Thank you very much for your thoroughgoing review. All points of your review are discussed in the file.

Round 3
Reviewer 1 Report
I appreciate the improvements applied to the manuscript. I have some minor comments:
- I would also be careful with the answers to the reviewers. In comment 3, the authors mention:
“When a system moves (vibrates in this case), the (dynamical) force (spring force) F=k x changes because x is harmonic (periodic)”, in this case, the system is linear. In the next sentence, they mention: “F(x) is the key characteristic of a vibrating system. Displacement x causes variation of the force F and vice versa. F(x) curve is known as stiffness characteristic, that can be linear or nonlinear”. These sentences are not consistent with the previous one when the system has been mentioned to be linear.
- Regarding the answer of comment 7, again, according to Figure 1, the joint IMJ seems to be a hinge joint, but in the models, there seems not to be any rotational degree of freedom. Please, discuss it in the discussion section. Is this a limitation? If not, why?
- The authors improved the English usage in the manuscript, which is good. However, in the answers to reviewers, there are some sentences that are clearly not correct: “stress-strain dependence is talking over in the manuscript”, “We do not have to perform “expensive” tests on the human being because numerical research can go with help”. For future submissions, even the reviewer can understand what is the final message, I would be sure that all sentences are understandable and correct.
- I would place Figs 9 and 10 within the Results section.
- Is t1 dimensionless in your expressions? In expression 10, if t is in the numerator, in the units of seconds, for example, t1 should have units. Please, check it throughout the manuscript (e.g. line 223).
- In the caption of figure 9, I would mention “as a function of…” omega. Omega is dimensionless in this case? If so, please mention in the figure caption to what values are normalized, not only omega but also amplitude and time (in case this is normalized).
- In figure 10, the units of a3 are not the same as in the other plots, which seems a bit weird having similar absolute values.
- The authors introduced: “Usually, the relaxation time of tissues is shorter than 0.5s.”, based on what? Can you provide a reference?
- The same for “the shift of the second resonance can be observed in real middle ear systems.”
- Line 256. Please correct the sentence.
- Line 265. I think that the same trials were used to track the parameters and to test the output, right? If so, this is not a validation, where other trials would be used to validate the results. I would say that “the fitted parameters can track the experimental data”.
Author Response
Authors would like to thank the reviewer very much for the very vigilant and important feedback that force us to bring the quality of the paper to the highest level. All changes, suggested by the reviewer, are marked in red colour in the manuscript, as usually.
